# *Lactiplantibacillus plantarum* dfa1 Outperforms *Enterococcus faecium* dfa1 on Anti-Obesity in High Fat-Induced Obesity Mice Possibly through the Differences in Gut Dysbiosis Attenuation, despite the Similar Anti-Inflammatory Properties

**DOI:** 10.3390/nu14010080

**Published:** 2021-12-25

**Authors:** Thunnicha Ondee, Krit Pongpirul, Kantima Janchot, Suthicha Kanacharoen, Thanapat Lertmongkolaksorn, Lampet Wongsaroj, Naraporn Somboonna, Natharin Ngamwongsatit, Asada Leelahavanichkul

**Affiliations:** 1Department of Preventive and Social Medicine, Faculty of Medicine, Chulalongkorn University, Bangkok 10330, Thailand; thunnichaon@yahoo.com (T.O.); kantima.janchot@gmail.com (K.J.); 2Department of International Health, Johns Hopkins Bloomberg School of Public Health, Baltimore, MD 21205, USA; 3Bumrungrad International Hospital, Bangkok 10110, Thailand; 4Department of Biology, Krieger School of Arts and Sciences, Johns Hopkins University, Baltimore, MD 21218, USA; jessiekanacharoen@gmail.com; 5Research Management and Development Division, Office of the President, Mahidol University, Nakhon Pathom 73170, Thailand; thanapat.ler@mahidol.edu; 6Department of Microbiology, Faculty of Science, Chulalongkorn University, Bangkok 10330, Thailand; lwongsaroj@gmail.com (L.W.); Naraporn.S@chula.ac.th (N.S.); 7Microbiome Research Unit for Probiotics in Food and Cosmetics, Chulalongkorn University, Bangkok 10330, Thailand; 8Department of Clinical Sciences and Public Health, Faculty of Veterinary Science, Mahidol University, Nakhon Pathom 73170, Thailand; natharin.nga@mahidol.edu; 9Department of Microbiology, Faculty of Medicine, Chulalongkorn University, Bangkok 10330, Thailand; 10Translational Research in Inflammation and Immunology Research Unit (TRIRU), Department of Microbiology, Chulalongkorn University, Bangkok 10330, Thailand; 11Nephrology Unit, Department of Medicine, Faculty of Medicine, Chulalongkorn University, Bangkok 10330, Thailand

**Keywords:** probiotics, dysbiosis, obesity, high fat diet, *Lactiplantibacillus plantarum*, *Enterococcus faecium*, *Lactobacillus plantarum*

## Abstract

Fat reduction and anti-inflammation are commonly claimed properties of probiotics. *Lactiplantibacillus plantarum* and *Enterococcus faecium* were tested in high fat-induced obesity mice and in vitro experiments. After 16 weeks of probiotics, *L. plantarum* dfa1 outperforms *E. faecium* dfa1 on the anti-obesity property as indicated by body weight, regional fat accumulation, serum cholesterol, inflammatory cytokines (in blood and colon tissue), and gut barrier defect (FITC-dextran assay). With fecal microbiome analysis, *L. plantarum* dfa1 but not *E. faecium* dfa1 reduced fecal abundance of pathogenic Proteobacteria without an alteration in total Gram-negative bacteria when compared with non-probiotics obese mice. With palmitic acid induction, the condition media from both probiotics similarly attenuated supernatant IL-8, improved enterocyte integrity and down-regulated cholesterol absorption-associated genes in Caco-2 cell (an enterocyte cell line) and reduced supernatant cytokines (TNF-α and IL-6) with normalization of cell energy status (extracellular flux analysis) in bone-marrow-derived macrophages. Due to the anti-inflammatory effect of the condition media of both probiotics on palmitic acid-activated enterocytes was neutralized by amylase, the active anti-inflammatory molecules might, partly, be exopolysaccharides. As *L. plantarum* dfa1 out-performed *E. faecium* dfa1 in anti-obesity property, possibly through the reduced fecal Proteobacteria, with a similar anti-inflammatory exopolysaccharide; *L. plantarum* is a potentially better option for anti-obesity than *E. faecium*.

## 1. Introduction

Obesity, a major health problem around the world, is linked to diabetes, dyslipidemia, and cardiovascular disease, all of which can cause serious consequences in critically ill patients [1]. Obesity-induced chronic inflammation leads to atherosclerosis, a major vascular consequence of obesity [2]. Despite the high prevalence of obesity-induced inflammation, the pathogenesis of this condition is still uncertain. It is thought to be a combination of several mechanisms, such as hypertrophic adipocyte hypoxia and apoptosis [3,4], reduced adiponectin with leptin elevation [5], saturated fatty acid-induced inflammation and mitochondria dysfunction [6], and metabolic endotoxemia from gut barrier defect [7]. Among these, the immune responses against endotoxins may lead to the most potent response because immune activation by the organism’s pathogen-associated molecular patterns (PAMPs) is naturally more severe than the response towards the host cell’s damage-associated molecular patterns (DAMPs) [8]. Perhaps, toll-like receptor 4 (TLR-4) activation by the lipid component, especially saturated fatty acid [9], might induce intestinal inflammation that leads to gut barrier defect and metabolic endotoxemia with systemic inflammation [10]. Indeed, endotoxin (lipopolysaccharide; LPS) has a molecular weight of 10–100 kDa and is found in the cell walls of Gram-negative bacteria, which are the most abundant organisms of gut microbiota [11]. The molecules with a molecular weight (MW) of more than 600 Da are unable to pass across the intestinal tight junction barrier under normal circumstances.

Obesity and a high-fat diet (HFD) produce gut dysbiosis [12] (an alteration of organisms in the intestine [13]), which increases gut-mucosal damage to the point where high-MW molecules, such as LPS, can be directly translocated into the liver and circulatory system [14,15]. Indeed, metabolic endotoxemia in obesity is well-known in both patients [14] and animal models [15]. Despite several animal models of obesity, diet-induced obesity mice are frequently used due to the cost-effectiveness and the substantial similarity to human obesity [14]. Likewise, non-alcoholic fatty liver disease (NAFLD), also called obesity-induced steatohepatitis, is a serious consequence of obesity that is exacerbated by the presence of LPS in blood circulation [16]. However, gut-leakage in obesity [14] is attenuated by host-beneficial probiotics [17,18,19], partly through the improved intestinal integrity by some anti-inflammatory substances [20,21,22,23,24]. Among all probiotic treatments against obesity, species *Lactiplantibacillus plantarum* and *Enterococcus faecium* are lactic acid-producing bacteria that are frequently used [25], partly due to the well-tolerance against acid and bile in the intestine [26] and the possible beneficial synergy with combined probiotics [27]. However, some strains of *E. faecium* may be pathogenically harmful to humans as causing several sites of infection (bacteremia, endocarditis, urinary tract infection) with the common anti-microbial resistance to many commonly used antibiotics [28]. Perhaps, *E. faecium* should be used if there is a prominent anti-obesity property and a comparison between the possible harmfulness versus the effectiveness of *E. faecium* are in need. On the other hand, the adverse effect of Lactobacilli probiotics is not prominent and is mostly reported in the immune-compromised host [29]. Recently, *L. plantarum* and *E. faecium* with the in vitro probiotic properties have been isolated from the Thai population and the Thai-isolated probiotics might be different from probiotics from the Caucasians due to the possible influence of some specific characteristics (ethnics, diets, climate, and co-evolution impact) [30,31]. Then, we hypothesized that there might be the differences in anti-obesity between strains of *L. plantarum* and *E. faecium* which could be used to determine the benefit of these probiotics. Additionally, the objective of the study was to initially compare the anti-obesity property of these interesting probiotics. Hence, *L. plantarum* dfa1 and *E. faecium* dfa1 were used in a mouse model with saturated fat-induced obesity and were also tested in the in vitro experiments.

## 2. Materials and Methods

### 2.1. Animals and Animal Model

The protocol of animal care and use was approved by the Institutional Animal Care and Use Committee of the Faculty of Medicine, Chulalongkorn University, Bangkok, Thailand (SST 025/2563) in compliance with USA National Institutes of Health standards. As such, 8-week-old male C57BL/6 mice were purchased from Nomura Siam (Pathumwan, Thailand). The regular mouse diet was a standard laboratory chow containing fat (4.5% *w*/*w*) with energy content calculated at 3.04 kcal/g (Mouse Feed Food No. 082, C.P. Company, Bangkok, Thailand), while the high-fat diet (HFD) containing fat, mostly from lard (60% *w*/*w*), with energy content calculated at 8.64 kcal/g [32] were used. A total of 32 mice were grouped by the block of four randomization into four experimental groups, including regular diet (RD), high fat diet (HFD), HFD with *L. plantarum* dfa1, and HFD with *E. faecium* dfa1. Average body weight at the randomization was 22.5 ± 0.5 g/mouse and separately kept in three cages (three mice/cage for two cages and a cage with two mice). The mice separation was performed for the fecal collection because the microbiome analysis from the same cage might be similar because of coprophagy (the consumption of feces from other mice). There were no death mice before the end of the experiments. Then, *L. plantarum* dfa1 or *E. faecium* dfa1 (Chulalongkorn University, Bangkok, Thailand) daily at 1 × 10^8^ colonies forming unit (CFU) in 0.5 mL phosphate buffer solution (PBS) or PBS alone for 16 weeks before sacrificing with cardiac puncture under isoflurane anesthesia. To determine the dose of probiotics, the spectrophotometer (Bio-Rad, Smart Spec 3000; Bio-Rad, Hercules, CA, USA) at optical density using 600 nm wavelength (OD600) of 0.1 (approximately 1 × 10^8^ CFU) in 0.5 mL PBS were performed. *L. plantarum* dfa1 and *E. faecium* dfa1 were isolated from the feces of volunteers participating in the RDG 6150124 project of Thailand Science Research and Innovation (TSRI) of the Faculty of Medicine, Chulalongkorn University; their functional properties were studied in vitro. Two probiotics were stored at −80 °C in MRSc broth for *L. plantarum* dfa1 and BHI broth for *E. faecium* dfa1 with 20% glycerol. Notably, placebo was not used in the HFD group without probiotics and the parameters in some mice of the group were missing due to the inadequate samples or technical difficulties. The probiotics were propagated each in broth; *L. plantarum* dfa1 and *E. faecium* dfa1 using De Man, Rogosa and Sharpe broth (MRSc) and Brain Heart Infusion broth (BHI), respectively, for 48 h at 37 °C, using 1% inoculum, and then sub-cultured at 37 °C for 48 h before use in each experiment.

At sacrifice, livers and skin were snap-frozen in liquid nitrogen and kept at −80 °C before use. Feces from all parts of the colon were combined and collected for microbiome analysis and were used to measure fecal burdens of *L. plantarum* dfa1 or *E. faecium* dfa1 using polymerase chain reaction (PCR).

### 2.2. Gut Leakage Measurement

Gut permeability was determined by fluorescein isothiocyanate dextran (FITC-dextran) assay, endotoxemia, and immunofluorescent detection of a tight junction protein (zona occludens-1; ZO-1) following previous publications [33,34,35]. As such, FITC-dextran, a nonabsorbable molecule with 4.4 kDa molecular mass (Sigma-Aldrich, St. Louis, MO, USA) at 12.5 mg per mice was orally administered at 3 h before the detection of FITC-dextran in serum by Fluorospectrometer (NanoDrop 3300; ThermoFisher Scientific, Wilmington, DE, USA). Serum endotoxin (LPS) was measured by HEK-Blue LPS Detection (InvivoGen, San Diego, CA, USA) and the data were recorded as 0 when LPS values were less than 0.01 EU/mL because of the limited lower range of the standard curve.

### 2.3. Analysis of Mouse Samples from Blood, Organs, and Feces

After fasting for 12 h after free access to drinking water, lipid profiles were measured by the quantification assay for triglyceride (TG), total cholesterol (Sigma-Aldrich, St. Louis, MO, USA), low- and high-density lipoprotein cholesterol (LDL and HDL) (Crystal Chem Inc., Downers Grove, IL, USA). Liver damage and serum cytokines were determined by EnzyChrom Alanine Transaminase assay (EALT-100, BioAssay, Hayward, CA, USA) and enzyme-linked immunosorbent assays (ELISA) for mouse cytokines (Invitrogen, Carlsbad, CA, USA), respectively. For histology, paraffin-embedded sections (4 µm thick) stained by Hematoxylin and Eosin (H&E) from 10% formalin-fixed samples were evaluated. The scoring system of obesity-induced liver damage was used like the following; steatosis (0–3), lobular inflammation (0–3), and hepatocellular ballooning degeneration (0–2) [36]. The thickness of subcutaneous fat was determined following a previous publication [37]. For the detection of lipids in the liver, livers were sonicated (High-Intensity Ultrasonic Processor, Newtown, CT, USA) in 500 µL of ice-cold PBS containing protease inhibitor Cocktail (I3786) (Sigma-Aldrich, St. Louis, MO, USA) and measured lipids from the supernatant by the quantification assay for triglyceride and total cholesterol (Sigma-Aldrich, St. Louis, MO, USA).

In addition, oxidative stress in the liver was evaluated following a previous study [38]. Briefly, livers were homogenized in radioimmunoprecipitation assay (RIPA) with protease inhibitor Cocktail (I3786) (Sigma-Aldrich, St. Louis, MO, USA) on ice before measuring an oxidative stress molecule, malondialdehyde (MDA), (Cayman Chemical Company, Ann Arbor, MI, USA). For an antioxidant molecule, livers were sonicated in 2-(N-morpholino) ethanesulfonic acid (MES) buffer (Sigma-Aldrich, St. Louis, MO, USA) before the measurement of glutathione (GSH) (Cayman Chemical Company, Ann Arbor, MI, USA) from the supernatant. Moreover, for cytokine detection in colon tissue, the samples were weighed, cut, thoroughly sonicated (High-Intensity Ultrasonic Processor, Newtown, CT, USA) in 500 mL of ice-cold PBS containing protease inhibitor Cocktail (I3786; Sigma-Aldrich, St. Louis, MO, USA) and measured cytokines from the supernatant by ELISA (Invitrogen; ThermoFisher Scientific, Wilmington, DE, USA).

### 2.4. Mouse Fecal Analysis

The bacterial abundance of *L. plantarum* dfa1 or *E. faecium* dfa1 was evaluated by real-time polymerase chain reaction (PCR) following a previous publication10. Briefly, total DNA was extracted by a QIAamp fast DNA Stool Mini Kit (Qiagen, Hiden, Germany), as per the manufacturer’s instructions. Primers for the variable regions of the 16 s ribosomal RNA (rRNA) gene sequence of *L. plantarum* dfa1 bacteria [39] were: Lp-F, 5′-AAAATCATGCGTGCGGGTAC-3′; Lp-R, 5′-ATGTTGCGTTGGCTTCGTCT-3′and for *E. faecium* dfa1 [40] were E16S 72f, 5′-CCGAGTGCTTGCACTCAATTGG-3′; E16S 210r, 5′-CTCTTATGCCATGCGGCATAAAC-3′. The constructive standard curve of *L. plantarum* dfa1 and *E. faecium* dfa1 was generated by the QuantStudio™ Design & Analysis Software v1.4.3 (Thermo Fisher Scientific, Wilmington, DE, USA) using 10-fold serial dilution with bacterial concentrations ranging from 2 to 2 × 10^5^ bacteria were used to convert the PCR results into bacterial copy numbers.

### 2.5. Fecal Microbiome Analysis

Feces from nine mice (0.25 g per mouse) from different cages in each experimental group were divided into three samples per group (three mice per sample) before performing microbiota analysis following a previous protocol [41]. In short, metagenomic DNA was extracted from 0.25 g feces by DNeasy PowerSoil Kit (Qiagen, Germantown, MD, USA). The Universal prokaryotic 515F (forward; (5′-GTGCCAGCMGCCGCGGTAA-3′) and 806R (reverse; 5′-GGACTACHVGGGTWTCTAAT-3′), with appended Illumina adapter and Golay barcode sequences, were used for 16S rRNA gene V4 library construction and sequenced using Miseq300 platform (Illumina, San Diego, CA, USA) at Omics Sciences and Bioinformatics Center, and Microbiome Research Unit for Probiotics in Food and Cosmetics, Chulalongkorn University. Raw sequences were quality processed and operational taxonomic unit (OTU) classified following Mothur’s standard operating platform procedures [42,43]. Bioinformatic analyses included good’s coverage, alpha diversity (e.g., Chao), and beta diversity (e.g., non-metric multidimensional scaling (NMDS)). Linear discriminant analysis effect size (LEfSe) and meta-stats were also performed to determine species marker and unique representing species of the interested group, respectively [42,44].

### 2.6. Anti-Inflammatory Responses of Enterocytes

Due to the possible differential impact of *L. plantarum* dfa1 versus *E. faecium* dfa1 on enterocytes, the in vitro experiments using the Caco-2 enterocyte cell line were performed. Caco-2 (ATCC HTB-37) from the American Type Culture Collection (Manassas, VA, USA) were maintained in supplemented Dulbecco’s modified Eagle medium (DMEM) at 37 °C under 5% CO_2_ and sub-cultured before use in the experiments. Then, Caco-2 cells at 1 × 10^6^ cells/well were incubated with palmitic acid at 0.5 mM/well or 100 ng/mL of lipopolysaccharide (LPS) from *E. coli* O26:B6 (Sigma-Aldrich, St. Louis, MO, USA) or palmitic acid with LPS (palmitic acid + LPS) with or without 5% (*vol*/*vol*) condition media of probiotics (each strain) (the total volume was adjusted into 200 μL/well by the culture media) for 24 h before determination of supernatant IL-8 using ELISA (Quantikine Immunoassay; R&D Systems, Minneapolis, MN, USA). For the preparation of probiotics condition media, *L. plantarum* dfa1 or *E. faecium* dfa1 at an OD600 of 0.1 were incubated anaerobically for 48 h before supernatant collection by centrifugation and filtration (0.22 µm membrane filter) (Minisart; Sartorius Stedim Biotech GmbH, Göttingen, Germany). After that, the supernatant of the samples (500 µL) was concentrated by speed vacuum drying at 40 °C for 3 h (Savant Instruments, Farmingdale, NY, USA), resuspended in an equal volume of DMEM, and stored at −20 °C until use. Due to the different enriched media for *L. plantarum* dfa1 or *E. faecium* dfa1 using De Man, Rogosa and Sharpe broth (MRSc) and Brain Heart Infusion broth (BHI), respectively, the media were also used as control. Additionally, to explore molecular characteristics of the effective anti-inflammatory molecules in condition media from *L. plantarum* dfa1 (LpCM) or *E. faecium* dfa1 (EfCM) against saturated fatty acid-induced enterocyte inflammation, various enzymes including; (i) α- amylase (in 20 mM sodium acetate with 7 mM sodium chloride), (ii) lipase (in 50 mM Tris–HCl), and (iii) proteinase K (in 50 mM Tris–HCl) (Sigma-Aldrich, St. Louis, MO, USA) at 1 mg/mL were incubated with the 24 h-activated Caco-2 cells (1 × 104 cells/well) (palmitic acid with or without LPS) at 25 °C (for amylase) and 37 °C (for lipase and proteinase) for 6 h before inactivation by heating at 100 °C for 10 min and supernatant IL-8 measurement by ELISA (R&D Systems, Minneapolis, MN, USA).

### 2.7. Gene Expression of Cholesterol Absorption-Associated Molecules in Enterocytes

To explore the impact of the probiotic condition media on cholesterol absorption, the RNA from stimulated Caco-2 cells was prepared using FavorPrep Tissue total RNA purification Mini Kit (Favorgen Biotech Corp, Vienna, Austria) and cDNA was synthesized by cDNA Synthesis assay (Thermo Fisher Scientific, Wilmington, DE, USA) before the detection by SYBR green-based real-time PCR (Thermo Fisher Scientific, Wilmington, DE, USA). The oligonucleotide primers for the experiment were (i) NPC Intracellular Cholesterol transporter-1 (*NPC-1*); forward 5′ -TAT GGT CGC CCG AAG CA-3′ and reverse 5′-TGC GGT TGT TCT GGA AAT ACTG-3′, (ii) ATP Binding Cassette Subfamily G Member 5 (*ABCG5*); forward 5′-ACC CAA AGC AAG GAA CGG GAA-3′ and reverse 5′-CAG CGT TCA GCA TGC CTG TGT-3′, (iii) ATP Binding Cassette Subfamily G Member 8 (*ABCG8*); forward 5′-GGG TGA GCG CAG GAG AGT CAG-3′ and reverse 5′-TCA CGC TGC TTT CCA CAC AGG-3′ and (iv) beta-actin (*β-actin*; a house-keeping gene); forward 5′-CCT GGC ACC CAG CAC AAT-3′ and reverse 5′-GCC GAT CCA CAC GGA GTA CT-3′. The forward and reverse primers were mixed in equal proportions and used at a final concentration of 10 uM. Samples were analyzed with QuantStudio6 Flex Real-time PCR System (Thermo Fisher Scientific, Wilmington, DE, USA) and initially preheated at 95 °C for 2 min. Then, 45 PCR cycles were performed as follows: 95 °C for 1 s, 60 °C for 1 min. Melting curve profiling was performed at the end of each PCR process to confirm amplification of specific transcripts by the following steps: 95 °C for 15 s, cooling to 60 °C for 1 min, heating the sample to 95 °C for 30 s, and cooling to 60 °C for 15 s under continuous measurement of fluorescence. The results were demonstrated in terms of relative quantitation of the comparative threshold (delta-delta Ct) method (2^−ΔΔCt^) as normalized by *β-actin*.

### 2.8. Transepithelial Electrical Resistance (Teer) and Enterocyte Permeability

The integrity of monolayer enterocytes in different conditions was determined by TEER using Caco-2 cells. Caco-2 cells (ATCC HTB-37) at 5 × 10^4^ cells per well were seeded onto the upper compartment of 24-well Boyden chamber trans-well plate using DMEM-high glucose supplemented with 20% Fetal Bovine Serum (FBS), 1% HEPES, 1% sodium pyruvate, and 1.3% Penicillin/Streptomycin for 15 days to establish the confluent monolayer. After that, palmitic acid at 0.5 mM/well with or without 5% (*vol*/*vol*) LpCM or EfCM were incubated at 37 °C under 5% CO_2_ for 24 h. Subsequently, TEER was measured by an epithelial volt-ohm meter (EVOM-2, World Precision Instruments, Sarasota, FL, USA) by placing the electrodes in the supernatant at the basolateral and apical chamber. The TEER value in media culture without cells was used as a blank and was subtracted from all measurements. The unit of TEER was ohm (Ω) × cm^2^. In parallel, 5 μL of FITC-dextran (4.4 kDa) (Sigma-Aldrich, St. Louis, MO, USA) at 10 mg/mL was added to the apical side of the trans-well chamber of the 24 h-stimulated Caco-2 cells. Then, FITC-dextran from the basolateral side of the trans-well plate was measured at 3 h after the incubation using Fluorospectrometer (NanoDrop 3300; ThermoFisher Scientific, Wilmington, DE, USA) as modified from the published protocols [45]. The concentration of FITC-dextran from the basolateral side represents the severity of permeability defect of Caco-2 cells.

### 2.9. Macrophage Cytokines and Extracellular Flux Analysis

Due to the influence of probiotics on gut inflammation [46,47,48] and the importance of macrophages on inflammation [49,50], an impact of condition media from *L. plantarum* dfa1 or *E. faecium* dfa1 were also tested in macrophages. Accordingly, bone marrow-derived macrophages were isolated from femurs and tibias of mice following a previous protocol [46]. Briefly, the bone marrow was collected by centrifugation at 6000 rpm for 4 °C and incubated for 7 days with DMEM supplemented with 10% fetal bovine serum (FBS), 1% penicillin/streptomycin, and 4-(2-hydroxyethyl)-1-piperazineethanesulfonic acid (HEPES) with sodium pyruvate in a humidified 5% CO_2_ incubator at 37 °C. Conditioned media of the L929 cell line, containing macrophage-colony stimulating factor, at 20% weight by volume (*w*/*v*), was used to induce macrophages from the pluripotent stem cells. Then, macrophages at 1 × 10^5^ cells/well were incubated for 24 h with the control media or 0.5 mM palmitic acid with or without 5% (*v*/*v*) of LpCM or EfCM before the determination of supernatant cytokines by ELISA (Invitrogen) or extracellular flux analysis using Seahorse XFp Analyzers (Agilent, Santa Clara, CA, USA) with oxygen consumption rate (OCR) and extracellular acidification rate (ECAR) representing mitochondrial function (respiration) and glycolysis activity, respectively. The stimulated macrophages at 1 × 10^5^ cells/well were incubated by Seahorse media (DMEM complemented with glucose, pyruvate, and L-glutamine) (Agilent, 103575-100) for 1 h before activation by different metabolic interference compounds, including oligomycin, carbonyl cyanide-4-(trifluoromethoxy)-phenylhydrazone (FCCP) and rotenone/antimycin A, for OCR evaluation. In parallel, glycolysis stress tests were performed using glucose, oligomycin, and 2-Deoxy-d-glucose (2-DG) for ECAR measurement. The data were analyzed by Seahorse Wave 2.6 software based on the following equations: (i) maximal respiration = OCR between FCCP and rotenone/antimycin A—OCR after rotenone/antimycin A; (ii) maximal glycolysis (glycolysis capacity) = ECAR between oligomycin and 2-DG—ECAR after 2-DG.

### 2.10. Statistical Analysis

Mean ± standard error (SE) was used for data presentation. The differences between groups were examined for statistical significance by one-way analysis of variance (ANOVA) followed by Tukey’s analysis or Student’s *t*-test for comparisons of multiple groups or 2 groups, respectively. All statistical analyses were performed with SPSS 11.5 software (SPSS Inc, Chicago, IL, USA) and Graph Pad Prism version 7.0 software (La Jolla, CA, USA). A *p*-value of <0.05 was considered statistically significant.

## 3. Results

### 3.1. Lactiplantibacillus plantarum Outperformed Enterococcus faecium in Obesity Attenuation in a High-Fat Diet Mouse Model

Both *L. plantarum* dfa1 and *E. faecium* dfa1 attenuated obesity in high-fat diet (HFD) mice as determined by body weight, serum lipid profile (total cholesterol and triglyceride), and visceral fat deposition in several sites (mesentery, peri-renal, retro-peritoneum, peri-gonadal, and subcutaneous fat) along with liver injury (liver weight, histological score and liver cholesterol) (Figure 1A–L) supported several studies [51,52]. In addition, both probiotics also attenuated liver enzyme (alanine transaminase), oxidative injury (reduced MDA and increased GSH), serum cytokines (TNF-α, IL-6, and IL-10), and gut leakage (FITC-dextran assay and endotoxemia) (Figure 2A–K). However, the anti-obesity effect of *L. plantarum* dfa1 was more profound than *E. faecium* dfa1 as indicated by body weight, blood cholesterol, some regional fat deposition (mesentery, peri-renal, subcutaneous fat, and liver), oxidative stress in the liver, serum pro-inflammatory cytokines (TNF-α and IL-6), and gut leakage (FITC-dextran assay) (Figure 1A–L or Figure 2A–K) without the difference in fecal abundance between both strains of probiotics (Figure 2L). These data supported the acceptable property of both probiotic strains in terms of the stability in the intestines (acid and bile tolerance) [26] and the possible difference in anti-obesity effect among probiotics [53]. Because (i) Gram-negative bacteria in the gut is a source of endotoxin (LPS) [11] which could enter blood circulation (obesity-induced endotoxemia) [54] and (ii) gut dysbiosis causes gut barrier defect is well-known [41,46,55], the better effect on gut barrier defect attenuation (serum FITC-dextran assay) of *L. plantarum* dfa1 when compared with *E. faecium* dfa1 might be due to the different effect on gut dysbiosis.

### 3.2. Lactiplantibacillus plantarum, but Not Enterococcus faecium, Reduced Proteobacteria (A Group of Pathogenic Bacteria) in Feces of High-Fat Diet Mice

In comparison with the regular diet group, HFD increased Proteobacteria (the pathogenic Gram-negative bacteria) without causing a difference in *Bacteroides* (the most abundance Gram-negative bacteria in feces) and Firmicutes (the high abundance bacteria in healthy condition) with *Bacteriodes* spp. as the possibly unique bacteria in HFD by Linear discriminant Effect Size (LEfSe) analysis (Figure 3A–D) supporting obesity-enhanced Bacteroides bacteria as previously reported [56]. However, HFD did not increase the total Gram-negative bacteria in feces (Figure 3D) and the alpha diversity (Figure 3E) when compared with the regular diet group. Despite the similar Gram-negative bacterial burdens in feces of HFD compared with regular diet mice (Figure 3D), HFD induced endotoxemia (Figure 2K) implying an impact of gut dysbiosis on the intestinal permeability [57,58]. On the other hand, there was an alteration of fecal microbiome analysis with probiotic administration. In microbiome analysis on the phylum level, *E. faecium* dfa1 did not alter the fecal abundance of Firmicutes, and Proteobacteria but reduced *Bacteroides*, while *L. plantarum* dfa1 reduced both Firmicutes and Proteobacteria but increased *Bacteroides* when compared with HFD (Figure 3A–D). In comparison between *L. plantarum* dfa1 and *E. faecium* dfa1 administration in HFD mice, *L. plantarum* dfa1 reduced Firmicutes and Proteobacteria but increased *Bacteroides* without the difference on total Gram-negative bacteria in feces (Figure 3D). The reduced Proteobacteria (pathogenic bacteria) in HFD mice after *L. plantarum* dfa1 administration (Figure 3D) might be, at least in part, responsible for the more potent weight reduction of *L. plantarum* dfa1 over *E. faecium* dfa1 (Figure 1A).

### 3.3. Both Lactiplantibacillus plantarum and Enterococcus faecium Attenuated Fatty Acid-Induced Enterocyte Inflammation through the Production of Carbohydrate Molecules

Because direct activation of saturated fatty acid and probiotic-producing molecules on enterocytes is possible [59,60], in vitro tests using palmitic acid (a representative saturated fatty acid) activation on enterocytes with or without the condition media from probiotics were performed. Indeed, the saturated fatty acid-induced supernatant cytokine (IL-8) and induced enterocyte tight junction injury, as indicated by TEER and transepithelial FITC-dextran (Figure 4A–C), with the up-regulation of several genes that associated with cholesterol absorption (*NPC-1*, *ABCG5*, and *ABCG8*) (Figure 4D–F). However, the condition media of both probiotics attenuated fatty acid-induced enterocyte injury, as indicated by reduced IL-8 production, and improved enterocyte integrity (TEER and transepithelial FITC-dextran) (Figure 4A–C). Likewise, the condition media also down-regulated the cholesterol absorption-associated genes (Figure 4D–F). Given that the synthesis of some anti-inflammatory molecules from probiotics might be, at least in part, responsible for the attenuation in gut barrier defect [61], enzyme neutralization experiments were performed. With the activation by saturated fatty acid with or without LPS, amylase neutralized the anti-inflammatory effect of condition media of *L. plantarum* dfa1, while both amylase and lipase neutralized the effect of *E. faecium* dfa1 (Figure 5A–F), implying that the characteristics of anti-inflammatory molecules of *L. plantarum* dfa1 and *E. faecium* dfa1 were carbohydrate and lipo-carbohydrate, respectively. The difference in active molecules of *L. plantarum* dfa1 and *E. faecium* dfa1 might, partly, be a factor that is responsible for the difference in anti-obesity potency.

However, these molecules from condition media of both probiotics similarly attenuated the pro-inflammatory effect of palmitic acid (saturated fatty acid) induced on macrophages as indicated by the reduction in supernatant cytokines (TNF-α and IL-6 but not IL-10) and macrophage cell energy status (Figure 6A–I). While palmitic acid-enhanced glycolysis activity (basal and maximal glycolysis) and reduced mitochondrial functions (basal and maximal respiration), the conditioned media shifted the responses toward the characteristics of the control group (Figure 6D–I). Because of the prominent use of glycolysis on cytokine production in macrophages [62], the reduced glycolysis and supernatant cytokines by condition media imply an impact of anti-inflammatory molecules from both probiotics.

## 4. Discussion

Although both *L. plantarum* dfa1 and *E. faecium* dfa1 attenuated obesity in high-fat diet (HFD)-administered mice, *L. plantarum* dfa1 out-performed *E. faecium* dfa1 as indicated by weight reduction, serum cytokines, and gut barrier defect, possibly through the more prominent reduction in fecal Proteobacteria with differences in the active anti-inflammatory substances.

### 4.1. Dysbiosis and Gut Barrier Defect in Obese Mice

The obese mice were overweight, increased fat accumulation, hyperlipidemia, liver injury (liver weight, steatohepatitis, and elevated liver enzymes), and gut barrier defect (FITC-dextran assay and increased serum LPS). Gut barrier defect causes obesity-induced endotoxemia which is a fundamental activator leading to several complications [63], including liver injury and cardiovascular diseases [11], as inflammatory responses against pathogens molecules are stronger than the responses toward self-antigens [8]. As such, HFD itself increased fecal pH, possibly through bile-production amplification, that reduces short-chain fatty acids (SCFAs) [64], increase Proteobacteria (mucosal invasive Gram-negative organisms [65,66]) but not Bacteroidetes (most prominent Gram-negative anaerobes in gut [67]) nor fecal burdens of total Gram-negative bacteria. This data supports that HFD enhances endotoxemia [68] through dysbiosis-induced intestinal mucosal injury [69,70], but does not increase PS burdens in the gut. Hence, the attenuation of dysbiosis and/or gut-leakage may be a direct adjunctive treatment against obesity-induced inflammation and other complications. Of note, the limitations of diet-induced obesity mice are poor standardization, long duration, and overtly obese when compared with the patients, although there are several similarities between mice and patients, including the obesity characteristics and insulin resistance [68]. Despites these limitations, the probiotics were tested with the same models.

### 4.2. Both Lactiplantibacillus plantarum and Enterococcus faecium Attenuated Obesity and Gut Dysbiosis

The attenuation of obesity-induced gut dysbiosis by probiotics [54,55,71,72] is explained through several mechanisms including the enhanced and effective energy in the host, reduced lipid absorption, increased SCFAs production, and promoted intestinal hormones [73]. Among several probiotics, *L. plantarum* dfa1 and *E. faecium* dfa1 demonstrate robust lactic acid production [26] that may alter HFD-induced dysbiosis. However, there was a different effect on fecal microbiome analysis between these probiotics. There was a more prominent reduction of Proteobacteria and Firmicutes with a higher abundance of *Bacteroides* after *L. plantarum* dfa1 administration when compared with *E. faecium* dfa1. As such, Proteobacteria is a major phylum of pathogenic Gram-negative bacteria, such as *Escherichia*, *Salmonella*, *Vibrio*, *Helicobacter*, and others [71] which might lead to the more severe gut barrier defect. The reduced Proteobacteria is one of the parameters indicating a more prominent beneficial effect of *L. plantarum* dfa1 over *E. faecium* dfa1. However, the abundance of Bacteroides after treatment with both probiotics was lower than the non-probiotics obese mice. *Bacteroides* are the most prominent Gram-negative bacteria in feces with a possible adverse effect in gut in some conditions [67], therefore the decreased *Bacteroides* supported probiotic advantage against obesity [10]. Although *L. plantarum* dfa1 reduced both beneficial bacteria; Firmicutes bacteria (mostly Gram-positive bacteria that prominently identified in healthy gut [74]), and pathogenic bacteria (Proteobacteria) in feces, the improved intestinal mucosal integrity indicates favorable outcomes. Indeed, gut barrier defect of obese mice with *L. plantarum* dfa1 was less severe than *E. faecium* dfa1 as determined by FITC-dextran assay. However, both probiotics were effectively reduced endotoxemia levels when compared with obese mice control and the level of endotoxin was not different between probiotics. The prominent reduction in serum FITC-dextran in *L. plantarum*-treated mice compared with *E. faecium*-administered mice with the non-difference in endotoxin levels, implying the higher sensitivity of FITC-dextran assay over LPS on gut barrier determination. Perhaps, this sensitivity difference is due to the smaller size of Dextran (4.4 kDa) when compared with LPS (>50 kDa) [11].

### 4.3. Both Lactiplantibacillus plantarum and Enterococcus faecium Attenuated Saturated Fatty Acid-Induced Inflammation in Enterocytes and Macrophages

Although the benefit on attenuation of gut dysbiosis (reduced Proteobacteria) by *L. plantarum* dfa1 was more potent than *E. faecium* dfa1 (possibly through the excretion of different active molecules), both probiotics attenuated obesity-induced intestinal inflammation as indicated by the lower cytokines from colon tissue compared with the non-probiotics group. To explore the molecular nature of immunomodulating substances in a conditioned medium against enterocytes, the enzyme neutralizing protocols on supernatant IL-8 production were used because of the predominant supernatant IL-8 in Caco-2 cells [75]. As such, the anti-inflammatory property of the condition media from *L. plantarum* dfa1 and *E. faecium* dfa1 was neutralized only by amylase enzyme and by amylase and lipase enzymes, respectively, indicating the active molecules with polysaccharide and lipo-polysaccharide, respectively. The anti-inflammatory effect on enterocytes improved enterocyte integrity (TEER) which might be responsible for the less severe obesity-enhanced gut barrier defect in mice. Additionally, the anti-inflammatory effect of the media from both probiotics was not only against palmitic acid but also toward LPS and LPS plus palmitic acid, implying a broad effect against several stimuli which might be suitable for medication. Although purification of bioactive substances was not performed, our initial characterization supposes the well-known importance of the exopolysaccharide from probiotics on anti-inflammatory effect [21,76,77]. In parallel, the probiotic molecules also down-regulated the cholesterol absorption associated molecules in similar to several probiotic studies [78].

Additionally, the molecules from condition media from both probiotics were also similarly attenuated palmitic acid-induced macrophage inflammation possibly through an alteration in cell energy status. Among all regions in the body, the intestine is the largest pool of macrophages which play a critical role in intestinal inflammation [79] and the attenuation of macrophage pro-inflammatory responses might be responsible for the less severe obesity-induced gut inflammation after probiotics administration. Here, the condition media from both probiotics down-regulate glycolysis activity, the main energy utilization for cytokine production [80]. Although the correlation between glycolysis and exopolysaccharide production from bacteria and the direct impact of probiotics on host enterocyte is mentioned [81,82], data on the direct influence of exopolysaccharide against host cell energy status are still less and the extraction of exopolysaccharide for the new anti-inflammatory drug is possible. With the “proof of concept” characteristic of the study, there were several limitations, especially on the mechanistic interpretation of the observed results. Further experiments on metagenomic, metabolomic, functional microbiota analysis, the correlation between bacterial abundance and observed anti-inflammatory effects and gut barrier integrity improvement are interesting. More studies on these topics are warranted for the future clinical translation.

## 5. Conclusions

*L. plantarum* dfa1 showed a more potent anti-obesity property than *E. faecium* dfa1 possibly through the pre-dominant attenuation on dysbiosis (reduction in pathogenic Proteobacteria in feces) and gut barrier defect. Additionally, *L. plantarum*, but not *E. faecium*, have the qualified presumption of safety (QPS) status by the European Food Safety Authority (EFSA) for the intentionally utilization in food. Although both probiotics attenuated inflammation in enterocytes and macrophages, possibly through exopolysaccharide, that is interesting to use as a new anti-inflammatory treatment, *L. plantarum* might be more proper for use for anti-obesity concerning possible pathogenesis of *E. faecium*.

## Figures and Tables

**Figure 1 nutrients-14-00080-f001:**
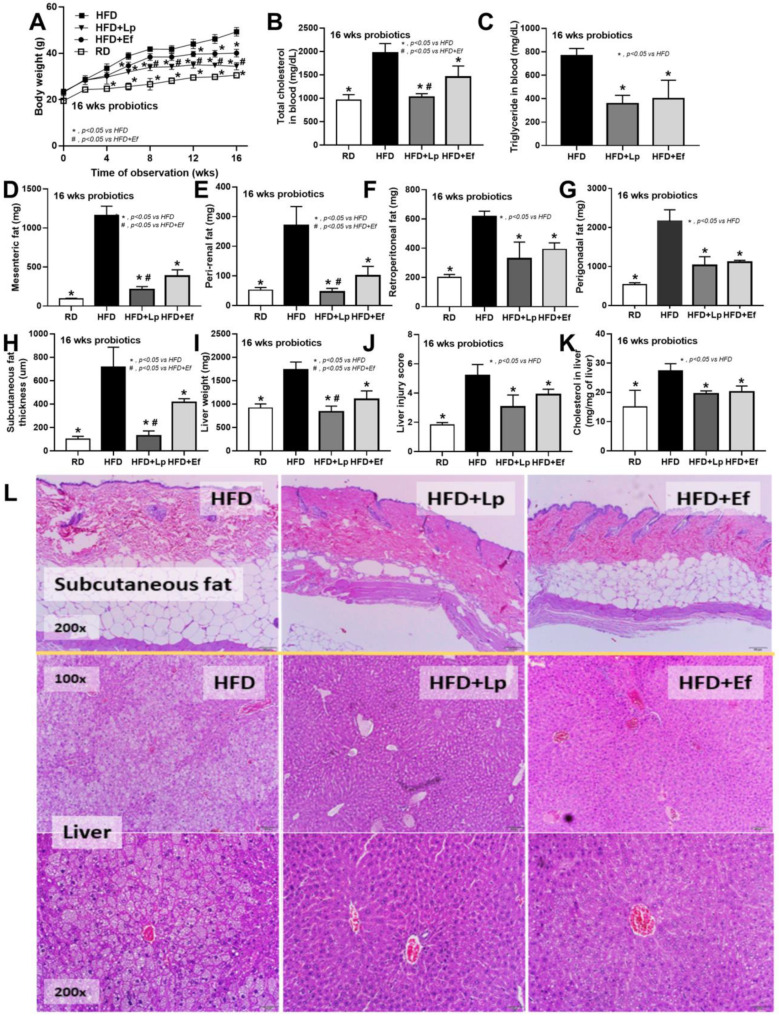
Characteristics of mice with regular diet (RD) or high fat diet (HFD) with or without *Lactiplantibacillus plantarum* (Lp) or *Enterococcus faecium* (Ef) as determined by body weight (**A**), fasting blood lipid profile (total cholesterol and triglyceride) (**B**,**C**), adipose tissue depots in several sites (**D**–**I**), subcutaneous fat thickness (**H**), liver injury (weight, histological score and cholesterol in liver) (**I**–**K**) and the representative figures of subcutaneous fat thickness and liver (original magnification 200×) (**L**) were demonstrated (*n* = 6–8/time-point or group).

**Figure 2 nutrients-14-00080-f002:**
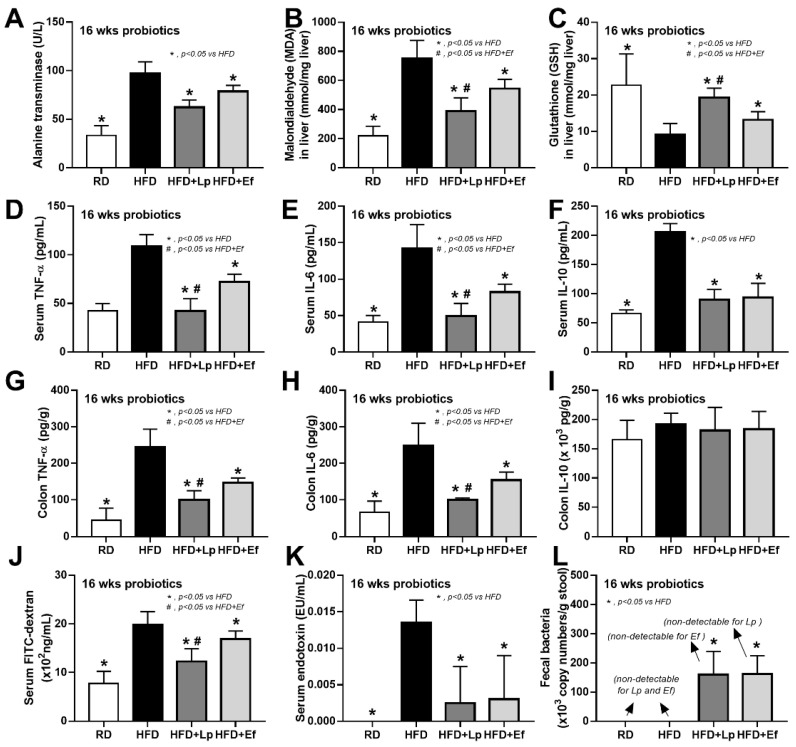
Characteristics of mice with regular diet (RD) or high fat diet (HFD) with or without *Lactiplantibacillus plantarum* (Lp) or *Enterococcus faecium* (Ef) as determined by alanine transaminase (ALT) (**A**), oxidative stress (malondialdehyde; MDA) and anti-oxidant molecule (glutathione) in liver (**B**,**C**), serum cytokines (TNF-α, IL-6 and IL-10) (**D**–**I**), gut leakage (FITC-dextran) (**J**), serum endotoxin (**K**) and bacterial abundance in feces by polymerase chain reaction (**L**) are demonstrated (*n* = 6–8/group).

**Figure 3 nutrients-14-00080-f003:**
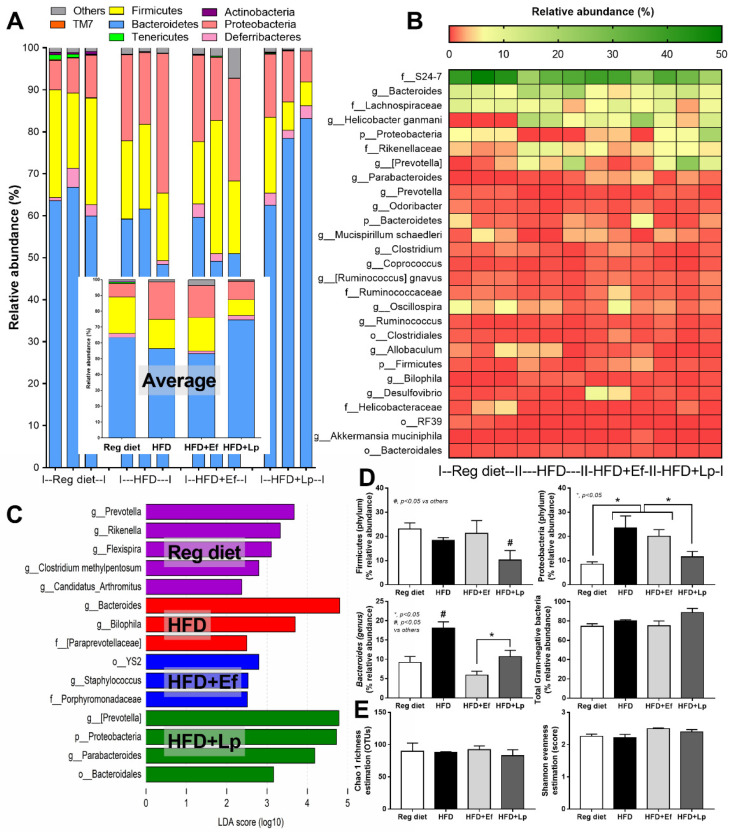
Gut microbiota analysis from feces of high fat diet (HFD) mice with or without *Lactiplantibacillus plantarum* (Lp) or *Enterococcus faecium* (Ef) or mice with regular diet (RD) as determined by the relative abundance of bacterial diversity at phylum (the inset graph is the average abundance) (**A**), and at genus (heat-map) (**B**), the possibly unique bacteria in each group by Linear discriminant Effect Size (LEfSe) analysis (**C**), graph presentation of the abundance of some groups of bacteria and total Gram-negative bacteria in feces (**D**) with the alpha diversity by Chao 1 richness estimation and Shannon evenness analysis (**E**) are demonstrated.

**Figure 4 nutrients-14-00080-f004:**
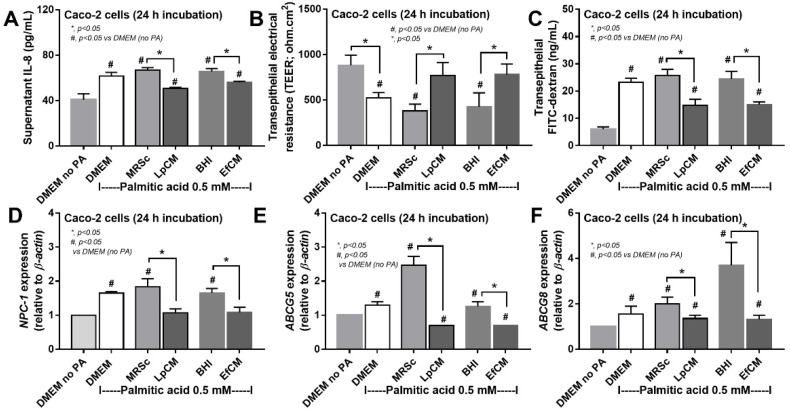
Characteristics of Caco-2 cells (enterocytes) with 0.5 mM palmitic acid (PA) with or without condition media of *Lactiplantibacillus plantarum* (LpCM) or *Enterococcus faecium* (EfCM) as indicated by supernatant IL-8 (**A**), transepithelial electrical resistance (TEER) (**B**), transepithelial FITC-dextran (**C**) and gene expression of cholesterol absorption associated molecules, including *NPC-1* (NPC Intracellular Cholesterol Transporter 1), *ABCG5* (ATP Binding Cassette Subfamily G Member 5) and *ABCG8* (ATP Binding Cassette Subfamily G Member 8), (**D**–**F**) are demonstrated. Notably, MRSc (De Man, Rogosa and Sharpe broth) and) and BHI (Brain Heart Infusion broth) were the culture media for Lp and Ef, respectively, and independent triplicate experiments were performed for all experiments.

**Figure 5 nutrients-14-00080-f005:**
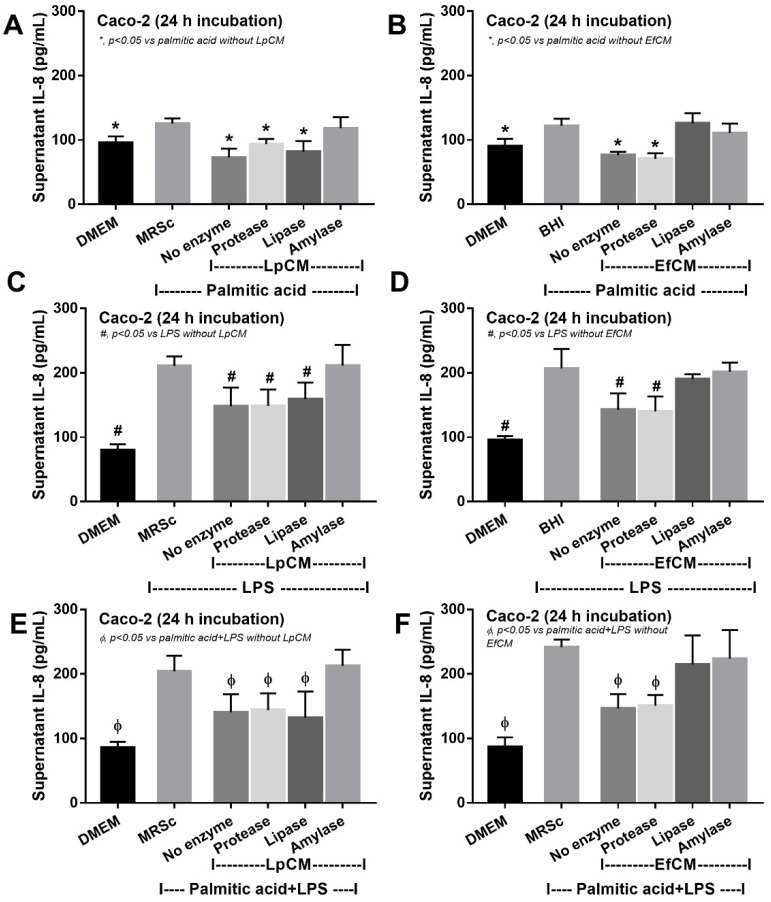
Supernatant IL-8 from Caco-2 cells (enterocytes) with Dulbecco’s Modified Eagle Medium (DMEM; media for Caco2 cell) or probiotic condition media from *Lactiplantibacillus plantarum* (LpCM) or *Enterococcus faecium* (EfCM) after incubated with or without enzyme inactivation (protease, lipase and amylase) when activated by 0.5 mM palmitic acid (**A**,**B**), 100 ng/mL of lipopolysaccharide (LPS) (**C**,**D**) or palmitic acid with LPS (**E**,**F**) are demonstrated. Notably, MRSc (De Man, Rogosa and Sharpe broth) and) and BHI (Brain Heart Infusion broth) were the culture media for Lp and Ef, respectively, and independent triplicate experiments were performed for all experiments.

**Figure 6 nutrients-14-00080-f006:**
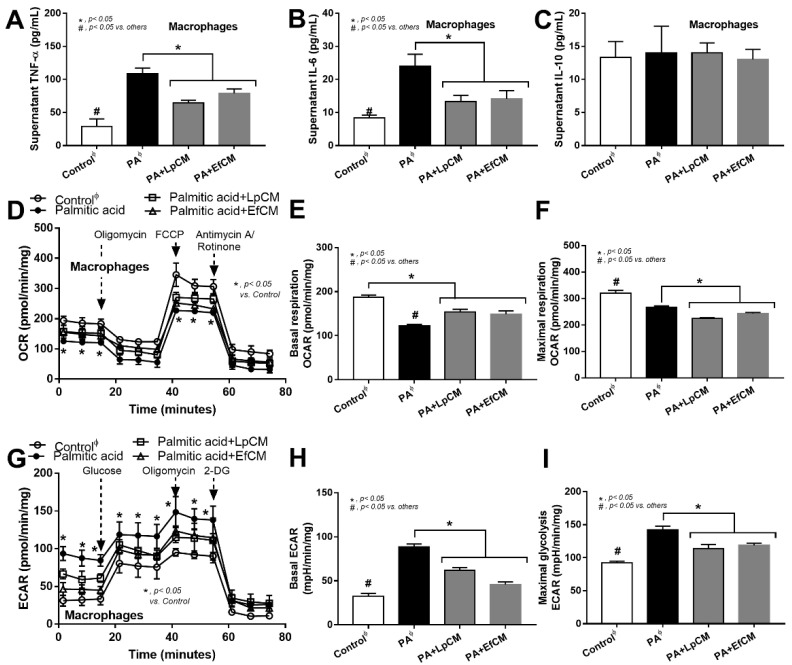
Characteristics of bone marrow-derived macrophages after 24 h activation in control or 0.5 mM palmitic acid with or without probiotic condition media from *Lactiplantibacillus plantarum* (LpCM) or *Enterococcus faecium* (EfCM) as indicated by supernatant cytokines (TNF-α, IL-6 and IL-10) (**A**–**C**) and extracellular flux analysis for mitochondrial activity, including oxygen consumption rate (OCR), basal respiration and maximal respiration, and glycolysis activity, including extracellular acidification rate (ECAR), basal ECAR, maximal glycolysis, (**D**–**I**) are demonstrated (independent triplicate experiments were performed for all experiments). Notably, ϕ indicates the combination of data form the activation by different control, including Dulbecco’s Modified Eagle Medium (DMEM; media for Caco2 cell) and bacterial culture media of Lp; MRSc (De Man, Rogosa and Sharpe broth) and for Ef; BHI (Brain Heart Infusion broth).

## Data Availability

Data is contained within the article.

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
