# Peer review of "Lactiplantibacillus plantarum dfa1 Outperforms Enterococcus faecium dfa1 on Anti-Obesity in High Fat-Induced Obesity Mice Possibly through the Differences in Gut Dysbiosis Attenuation, despite the Similar Anti-Inflammatory Properties"

_nutrients, 2021, doi:10.3390/nu14010080_

Round 1
Reviewer 1 Report
In their manuscript Ondee et al. describe the effect of two probiotic bacteria Lactobvacillus plantarum and Enterococcus faecium on obese mice. Both lead to weight reduction but L. plantarum outperforms the other strain. The authors then test many criteria in which L. plantarum might be superior including serum cytokines, gut barrier defects as well as bacterial microbiom compositions amongst others. The work is comprehensive and well described. the conclusions are very intersting.
Author Response
Reviewer 1: In their manuscript Ondee et al. describe the effect of two probiotic bacteria Lactobacillus plantarum and Enterococcus faecium on obese mice. Both lead to weight reduction but L. plantarum outperforms the other strain. The authors then test many criteria in which L. plantarum might be superior including serum cytokines, gut barrier defects as well as bacterial microbiom compositions amongst others. The work is comprehensive and well described. the conclusions are very interesting.
Response: Thank you very much for the compliments. We are pleased to know that our work is satisfactory.
Reviewer 2 Report
Dear Authors!
The concept of the study is very interesting and the issues investigated should become the subject of clinical study. At this point, the question arises: What is the practical translation of the animal model used for obesity in humans? The authors should address this question very carefully in the introduction and discussion. The next problem concerns a strain belonging to the species E. faecium, which does not have EFSA QPS status (Scientific Opinion on the update of the list of QPS-recommended biological agents intentionally added to food or feed as notified to EFSA (2017-2019). I do not think that the lack of this status should preclude research on strains of this species, but this issue should be thoroughly discussed in the paper.
Other major concerns:
1. The taxonomic name of Lactobacillus plantarum has been changed to Lactiplantibacillus plantarum. Please use the full strain name Lactiplantibacillus plantarum dfa1 or L. plantarum dfa1 throughout the paper, please give also the full strain name of E. faecium.
2) Please add: body weight of mice, method of randomisation, number of animals per group, drop outs number and reasons, did any animals died, how many animals were in the cage, how the probiotic was administered, whether a placebo was used.
(3) The authors should discuss the taxonomic changes in more depth in the context of the observed pathophysiological benefits. Readers should be able to find out if the observed changes in bacterial abundance are related to anti-inflammatory effects and improving intestinal barrier function (postulated mechanisms).
(4) The study has numerous limitations that have not been mentioned or discussed. These include: only two measurements, which makes mechanistic interpretation of the observed results difficult; no metagenomic analysis (only 16s method), no metabolomic analysis, no functional analysis of the microbiota, no analysis of the correlation between bacterial abundance and observed antinflammatory effects and gut barrier integrity improvement.
Minor:
For the sake of order, the objectives of the study and the research hypotheses should be listed.
Line 74 - after comma please add word "species".
Author Response
Reviewer 2: The concept of the study is very interesting and the issues investigated should become the subject of clinical study. At this point, the question arises: What is the practical translation of the animal model used for obesity in humans? The authors should address this question very carefully in the introduction and discussion.
Response: We are pleased to know that our work is well-received and agree that further clinical study is warranted. We are aware that there are several limitations and benefit aspects of the animal model; however, the same model was used for comparing the two probiotics. We added several sentences in the Introduction and Discussion as suggested. Introduction: “Indeed, metabolic endotoxemia in obesity is well-known in both patients [14] and animal models [15]. Despite several animal models of obesity, diet-induced obesity mice are frequently used due to the cost-effectiveness and the substantial similarity to human obesity [14].” Discussion: “Of note, the limitations of diet-induced obesity mice are poor standardization, long duration, and overtly obese when compared with the patients, although there are several similarities between mice and patients, including the obesity characteristics and insulin resistance [68]. Despites these limitations, the probiotics were tested with the same models.”
Reviewer 2: The next problem concerns a strain belonging to the species E. faecium, which does not have EFSA QPS status (Scientific Opinion on the update of the list of QPS-recommended biological agents intentionally added to food or feed as notified to EFSA (2017-2019). I do not think that the lack of this status should preclude research on strains of this species, but this issue should be thoroughly discussed in the paper.
Response: We thank the reviewer for the insightful opinion and useful information. Actually, we were not aware of this point and it seems that using E. faecium in foods is even more questionable. We add the following statement in the Conclusion of the manuscript: “Additionally, L. plantarum, but not E. faecium, have the qualified presumption of safety (QPS) status by the European Food Safety Authority (EFSA) for the intentionally utilization in food”.
Reviewer 2: The taxonomic name of Lactobacillus plantarum has been changed to Lactiplantibacillus plantarum. Please use the full strain name Lactiplantibacillus plantarum dfa1 or L. plantarum dfa1 throughout the paper, please give also the full strain name of E. faecium.
Response: The taxonomic names were updated as advised.
Reviewer 2: Please add: body weight of mice, method of randomization, number of animals per group, drop outs number and reasons, did any animals died, how many animals were in the cage, how the probiotic was administered, whether a placebo was used.
Response: Thank you very much for the comment. The following information was added accordingly: “A total of 32 mice were grouped by the block of four randomization in to 4 experimental groups, including regular diet (RD), high fat diet (HFD), HFD with L. plantarum dfa1, and HFD with E. faecium. Average body weight at the randomization was 22.5 ± 0.5 g/mouse and separately kept in 3 cages (3 mice/cage for 2 cages and a cage with 2 mice). The mice separation was performed for the fecal collection because the microbiome analysis from the same cage might be similar because of coprophagy (the consumption of feces from other mice). There were no death mice before the end of the experiments.” and “Notably, placebo was not used in the HFD group without probiotics and the parameters in some mice of the group were missing due to the inadequate samples or technical difficulties.”
Reviewer 2: The authors should discuss the taxonomic changes in more depth in the context of the observed pathophysiological benefits. Readers should be able to find out if the observed changes in bacterial abundance are related to anti-inflammatory effects and improving intestinal barrier function (postulated mechanisms).
Response: Thank you for the comment and we apologize for the unclear presentation. More explanation about the effect of the observed taxonomy was added to the Discussion on this topic as following “As such, Proteobacteria is a major phylum of pathogenic Gram-negative bacteria, such as Escherichia, Salmonella, Vibrio, Helicobacter, and others [71] which might lead to the more severe gut barrier defect. The reduced Proteobacteria is one of the parameters indicating a more prominent beneficial effect of L. plantarum over E. faecium. However, the abundance of Bacteroides after treatment with both probiotics was lower than the non-probiotics obese mice. Because Bacteroides are the most prominent Gram-negative bacteria in feces with a possible adverse effect in gut in some conditions [67], the de-creased Bacteroides supported probiotic advantage against obesity [10].”.
Reviewer 2: The study has numerous limitations that have not been mentioned or discussed. These include: only two measurements, which makes mechanistic interpretation of the observed results difficult; no metagenomic analysis (only 16s method), no metabolomic analysis, no functional analysis of the microbiota, no analysis of the correlation between bacterial abundance and observed anti-inflammatory effects and gut barrier integrity improvement.
Response: We thank the reviewer for the comments. These points were added as the limitations of the study: “With the “proof of concept” characteristic of the study, there were several limitations, especially on the mechanistic interpretation of the observed results. Further experiments on metagenomic, metabolomic, functional microbiota analysis, the correlation between bacterial abundance and observed anti-inflammatory effects and gut barrier integrity improvement are interesting. More studies on these topics are warranted for the future clinical translation.”
Reviewer 2: For the sake of order, the objectives of the study and the research hypotheses should be listed.
Response: The research objectives and hypotheses were orderly listed in the Introduction as suggested: “Then, we hypothesized that there might be the differences in anti-obesity between L. plantarum and E. faecium which could be used to determine the benefit of these probiotics. Also, the objective of the study was to initially compare the anti-obesity property of these interesting probiotics”.
Reviewer 2: Line 74 - after comma please add word "species".
Response: The word was added accordingly.
We hope that our responses are satisfactory. Thank you very much for the useful comments and kind consideration.
Round 2
Reviewer 2 Report
Dear Authors,
The manuscript was significantly improved, but doubts remain about the use of the obesity model chosen by the authors. In order not to mislead readers, I propose to modify the title of the paper as follows: Lactiplantibacillus plantarum dfa1 outperforms Enterococcus faecium dfa1 on anti-obesity possibly in high fat-induced obesity mice through the differences in gut dysbiosis attenuation, despite the similar anti-inflammatory properties.
I also propose to follow the comments below, so that the lack of order in this area does not disturb readers' perception of this interesting work.
Line 95: …between strains of…
Line 97: Hence Lactiplantibacillus plantarum dfa1 and Enterococcus faecium dfa1 were used…
After this sentence, the names of the strains should be unified throughout the text and the abstract. I propose to use the names: L. plantarum dfa1 and E. faecium dfa1
I cannot find technical information on how the probiotics were administered.
Author Response
Reviewer: The manuscript was significantly improved, but doubts remain about the use of the obesity model chosen by the authors. In order not to mislead readers, I propose to modify the title of the paper as follows: Lactiplantibacillus plantarum dfa1 outperforms Enterococcus faecium dfa1 on anti-obesity possibly in high fat-induced obesity mice through the differences in gut dysbiosis attenuation, despite the similar anti-inflammatory properties.
Response: We agree with the reviewer and correct it accordingly.
Reviewer: I also propose to follow the comments below, so that the lack of order in this area does not disturb readers' perception of this interesting work.
Line 95: …between strains of…
Line 97: Hence Lactiplantibacillus plantarum dfa1 and Enterococcus faecium dfa1 were used…
After this sentence, the names of the strains should be unified throughout the text and the abstract. I propose to use the names: L. plantarum dfa1 and E. faecium dfa1
Response: We thank the reviewer for the comment and correct them accordingly.
Reviewer: I cannot find technical information on how the probiotics were administered.
Response: We add it in the method section of the new version manuscript as following “To determine the dose of probiotics, the spectrophotometer (Bio-Rad, Smart Spec 3000; Bio-Rad, Hercules, CA, USA) at optical density using 600 nm wavelength (OD600) of 0.1 (approximately 1x108 CFU) in 0.5 ml PBS were performed.” and “The probiotics were propagated each in broth; L. plantarum dfa1 and E. faecium dfa1 using De Man, Rogosa and Sharpe broth (MRSc) and Brain Heart Infusion broth (BHI), respectively, for 4824 hours at 37 °C, using 1% inoculum, and then sub-cultured at 37 °C for 4824 hours before use in each experiment.”.